# Zinc Oxide Nanoparticles Induce Autophagy and Apoptosis via Oxidative Injury and Pro-Inflammatory Cytokines in Primary Astrocyte Cultures

**DOI:** 10.3390/nano9071043

**Published:** 2019-07-21

**Authors:** Woo-Ju Song, Myung-Seon Jeong, Dong-Min Choi, Kil-Nam Kim, Myung-Bok Wie

**Affiliations:** 1Department of Veterinary Toxicology, College of Veterinary Medicine and Institute of Veterinary Science, Kangwon National University, Chuncheon 24341, Korea; 2Chuncheon Center, Korean Basic Science Institute, Chuncheon 24341, Korea; 3Department of Biochemistry, Kangwon National University, Chuncheon 24341, Korea

**Keywords:** astrocyte cultures, autophagy, apoptosis, pro-inflammatory cytokines, zinc oxide nanoparticles

## Abstract

The present study examined the potential toxic concentrations of zinc oxide nanoparticles (ZnO NPs) and associated autophagy and apoptosis-related injuries in primary neocortical astrocyte cultures. Concentrations of ZnO NPs ≥3 μg/mL induced significant toxicity in the astrocytes. At 24 h after exposure to the ZnO NPs, transmission electron microscopy revealed swelling of the endoplasmic reticulum (ER) and increased numbers of autophagolysosomes in the cultured astrocytes, and increased levels of LC3 (microtubule-associated protein 1 light chain 3)-mediated autophagy were identified by flow cytometry. Apoptosis induced by ZnO NP exposure was confirmed by the elevation of caspase-3/7 activity and 4′,6′-diamidino-2-phenylindole (DAPI) staining. Significant (*p* < 0.05) changes in the levels of glutathione peroxidase, superoxide dismutase, tumor necrosis factor (TNF-α), and interleukin-6 were observed by enzyme-linked immunoassay (ELISA) assay following the exposure of astrocyte cultures to ZnO NPs. Phosphatidylinositol 3-kinase (PI3K)/mitogen-activated protein kinase (MAPK) dual activation was induced by ZnO NPs in a dose-dependent manner. Additionally, the Akt (protein kinase B) inhibitor BML257 and the mTOR (mammalian target of rapamycin) inhibitor rapamycin contributed to the survival of astrocytes. Inhibitors of cyclooxygenase-2 and lipoxygenase attenuated ZnO NP-induced toxicity. Calcium-modulating compounds, antioxidants, and zinc/iron chelators also decreased ZnO NP-induced toxicity. Together, these results suggest that ZnO NP-induced autophagy and apoptosis may be associated with oxidative stress and the inflammatory process in primary astrocyte cultures.

## 1. Introduction

Zinc oxide nanoparticles (ZnO NPs) have been widely used for medicinal purposes, such as photodynamic therapy [1] and piezoelectric nanomaterials for biomedical applications [2,3,4], various consumer products such as cosmetics and food additives, and industrial materials such as plastics, paints, and ceramics [5,6]. However, concerns of their potential toxicity in terms of environmental safety and human health problems have increased alongside the use of these nanoparticles [5,6,7]. Although ZnO NPs have some positive medical applications due to their antibacterial and anticancer effects, brain injury may occur at any time in both human and non-human biological systems because these nanoparticles can break down the blood–brain barrier (BBB) or be uptaken via olfactory sensory neurons in the nose following prolonged exposure or overdose [8,9,10]. Recently, there has been a gradual increase in neurotoxic and gliotoxic research on ZnO NPs aimed at identifying the most susceptible areas of the central nervous system (CNS) [11,12,13,14].

Zinc is one of several essential endogenous metallic ions, and is thought to be a modulator of critical physiological functions such as systemic growth, bone metabolism, wound healing, endocrine function, neuronal excitability, and survival in the brain [15,16]. However, zinc is also involved in a variety of neurodegenerative processes including Alzheimer’s disease, ischemic stroke, traumatic brain injury, epilepsy, depression, and aging [17,18]. Of the primary toxic mechanisms associated with ZnO NPs, oxidative stress and inflammation induced by the accumulation of free zinc ions released from ZnO NPs may be the most important components of its neuropathological etiology [10,19,20]. 

A recent study employing an ischemic stroke model found that zinc overload-induced activation and/or the cell death of astrocytes is correlated with autophagy [21,22] and apoptosis [13,23]. ZnO NPs increase the expression of caspase-3 and cyclooxygenase (COX)-2 in the cerebellum [24]. Additionally, the gliosis of astrocytes and expression of the pro-inflammatory cytokines interleukin (IL)-6 and tumor necrosis factor (TNF)-α all increase in the brain after ZnO NP treatment; at the same time, the antioxidant enzymes glutathione peroxidase (GPx) and superoxide dismutase (SOD) are also abruptly reduced, according to increasing zinc concentration [12,24]. Wang et al. [13] reported that ZnO NPs induce oxidative stress-mediated apoptosis via JNK(c-Jun N-terminal kinase )/ERK(extracellular signal-regulated kinase)/p38 mitogen-activated protein kinase (MAPK) signaling pathways in primary cultured astrocytes. Although the number of apoptosis and autophagy studies using ZnO NPs is increasing, few reports have investigated ZnO NP-related toxicity in astrocytes, which are glial cells that play important roles in the CNS [19,25]. Thus, the present study investigated the role of autophagy and apoptosis following exposure to ZnO NPs in primary astrocyte cultures. The influence of ZnO NPs on phosphatidylinositol 3-kinase (PI3K)/mitogen-activated protein kinase (MAPK) activation was also evaluated. An et al. [26] reported that zinc activates both the PI3K and MAPK pathways. Since astrocytes are considered an important regulator of survival or injury in neuronal cells, it would be valuable to examine autophagic or apoptotic changes following exposure to ZnO NPs. Moreover, various protective strategies against ZnO NP-mediated toxicity are necessary. 

Previously, we reported that lipoxygenase (LOX), not COX-2, is the predominant arachidonate enzyme in ZnO NP-related toxicity in human dopaminergic neuroblastoma SH-SY5Y cells [19]. However, we also determined that inhibitors of COX-2 and/or LOX exhibit protective effects against ZnO NP-induced toxicity in human mesenchymal stem cells [27]. Accordingly, it would be interesting to examine which is predominant, COX-2 or LOX, in the cytotoxicity induced by ZnO NPs in primary cultured astrocytes. Therefore, in this study, we investigated (1) significant toxic concentrations of ZnO NPs; (2) astrocyte morphology using transmission electron microscopy (TEM) and mean autophagy intensity using the LC3 (microtubule-associated protein 1 light chain 3) antibody; (3) caspase-3/7 activity and nuclear fluorescence staining; (4) the pro-inflammatory cytokines IL-6 and TNF-α, and antioxidant enzymes SOD and GPx; (5) PI3K/MAPK activation; and (6) the protective effects of various chemicals, such as COX-2/LOX, PI3K, Akt (protein kinase B), and mTOR (mammalian target of rapamycin) inhibitors, calcium modulators, antioxidants, and zinc/iron chelators that might protect astrocytes from ZnO NP-induced injury.

## 2. Materials and Methods

### 2.1. Chemicals and ZnO NP Suspension

ZnO NPs (Lot No.: D28X017, Cat. No. 44898, ZnO NPs, ZnO NanoGard^®^) were obtained from Alfa Aesar Co. (Ward Hill, Haverhill, MA, USA); the average particle size of the ZnO NPs in the powder was 67 nm (40–100 nm), and the range of the specific surface area (based on the Brunauer–Emmett–Teller [BET] theory) was 16 m^2^/g (10–25 m^2^/g). Horse serum (HS), fetal bovine serum (FBS), penicillin-streptomycin, and Minimal Essential Eagle’s Medium (MEM) were purchased from Gibco Co. (Grand Island, NY, USA); N,N,N′,N′-tetrakis-(2-pyridylmethyl) ethylenediamine (TPEN), LY294002, and rapamycin were purchased from Enzo Life Sciences (Farmingdale, NY, USA); and esculetin, meloxicam, phenidone, N-acetylcysteine (NAC), deferoxamine mesylate, 4′,6′-diamidino-2-phenylindole (DAPI), and 3-(4,5-dimethylthiazol-2-yl)-2,5-diphenyltetrazolium bromide (MTT) were purchased from Sigma Chemical Co. (St. Louis, MO, USA). The ZnO NP suspensions were prepared with Millipore water at a concentration of 1 mg/mL. Prior to adding the ZnO NPs to the culture media, each suspension was sonicated for at least 30 min and then vigorously vortexed. Thereafter, the suspension was immediately applied to the cultured astrocytes.

### 2.2. Astrocyte Cultures and ZnO NP Treatment

Murine astrocyte cultures were prepared as previously described [28,29]. Briefly, the astrocyte cultures were prepared from the neocortices of newborn mice (postnatal day 1–2) and plated at 1.5 hemispheres per plate in MEM supplemented with 2 mM of glutamine, 10% FBS, 10% HS, and 100 U penicillin/100 µg/mL of streptomycin. Then, the cells were incubated in a humidified atmosphere at 37 °C with 5% CO_2_, and cytosine arabinoside (final concentration, 10 µM) was added to halt the overgrowth of the astrocyte cultures and dividing cells. The culture medium was exchanged every 3 to 4 days until experiments were performed. The astrocyte cultures were maintained for 3 to 5 weeks, until the grown cells formed a confluent monolayer. The study protocol was approved by Kangwon National University Institutional Review Board (KW-170322-1 and KW-180621-1). 

### 2.3. Measurement of Lactate Dehydrogenase (LDH) Activity

After changing the growing media to MEM media without serum, the cultured astrocytes were treated with either 3 µg/mL or 5 µg/mL of ZnO NPs or co-treated with ZnO NPs and test chemicals for 20 to 24 h. After treatment, cell injury was quantitatively estimated at 20 to 24 h by measuring lactate dehydrogenase (LDH) release from damaged cells into the culture medium, as described previously [30]. LDH activity was measured at a wavelength of 340 nm using a kinetic program on a VersaMax Eliza Reader (Molecular Devices, San Jose, CA, USA).

### 2.4. MTT Assay

The astrocytes were cultured in a 96-well culture plate at 37 °C in a 5% CO_2_ incubator. After a monolayer of astrocytes was formed, the cells were exposed to ZnO NPs and/or test chemicals in 200-µL volume, and then incubated for 20 to 24 h. Next, the MTT solution was added, and the cells were incubated at 37 °C for 2 to 3 h. After discarding the culture medium, 100 µL of dimethyl sulfoxide (DMSO) was added, and the cells were incubated for 30 min to dissolve the formazan crystals. The viability of the cells was estimated at 570 nm using a VersaMax microplate reader (Molecular Devices). 

### 2.5. LC3-Antibody Detection

To assess autophagy, cultured astrocytes growing in six-well plates were exposed to ZnO NPs for 12 or 24 h. Autophagy Reagent A was applied to the cultured cells for 3 h; then, the cells were washed once with 1× Hank’s Balanced Sat Solution (HBSS) and accutase for 5 min at 37 °C to detach the cells. The harvested cells were transferred to Muse Assays microtubes, and 100 µL of 1× HBSS was added to each well. Next, the cells were centrifuged at 300× *g* for 5 min at 4 °C, the supernatant was removed, anti-LC3 Alexa Fluor^®^ 555 and 1× Autophagy Reagent B were added, and the samples were incubated on ice for 30 min in the dark. Next, the samples were centrifuged at 300× *g* for 5 min at 4 °C, the supernatant was removed, and then each sample was resuspended with 200 µL of 1× assay buffer and assayed using Muse^TM^ flow cytometry. 

### 2.6. Transmission Electron Microscopy (TEM)

Prior to TEM, the cultured astrocytes were washed with 0.1 M of phosphate-buffered saline (PBS) and fixed with a mixture of 1% paraformaldehyde and 4% glutaraldehyde overnight at 4 °C. Then, the astrocytes were post-fixed in 1% osmium tetroxide in the same buffer and dehydrated with ethanol and propylene oxide. Subsequently, the samples were embedded in Epon-812 resin, and ultra-thin sections were obtained using an ultra-cut microtome (Leica Co., Greenwood Village, CO, USA). Finally, the sections were stained with uranyl acetate and lead citrate and subjected to TEM visualization (LEO912AB, Carl Zeiss, Oberkochen, Germany).

### 2.7. ELISA Assay

To measure the activities of IL-6, TNF-α, SOD, and GPx, a mouse enzyme-linked immunoassay (ELISA) assay kit (CUSABIO, Hubei Province, China) was used according to the manufacturer’s instructions. Briefly, IL-6 and TNF-α activities were assessed by collecting 100 μL of media from cultured astrocytes. To assess the activities of SOD and GPx, cultured astrocytes were treated with Pierce^TM^ RIPA Buffer (Thermo Fisher Scientific, Waltham, MA, USA) and Halt^TM^ Protease Inhibitor Cocktail (Thermo Fisher Scientific), and sonicated in 3-min ON and 30-s OFF cycles. The cycle was repeated three times. After sonication, the cells were centrifuged at 12,000 rpm for 20 min at 4 °C, and 100 μL of supernatant was transferred to a new tube. Next, 100 μL of media (IL-6, TNF-α) or supernatant (SOD, GPx) and standard were added to specific antibody-coated 96-well microplates and incubated at 37 °C in a 5% CO_2_ incubator for 2 h. All of the supernatant was removed, 100 μL 1× biotin antibody was added to each well, and the plates were incubated at 37 °C in a 5% CO_2_ incubator for 1 h. Then, each well was washed three times with 1× wash buffer, 100 μL 1× HRP (horseradish peroxidase)-avidin was added, and the plates were incubated at 37 °C in a 5% CO_2_ incubator for 1 h. Each well was washed five times, 90 μL of TMB (3,3′,5,5′-tetramethylbenzidine) substrate was added, and the plates were incubated at 37 °C in a 5% CO_2_ incubator for 25 min in the dark. Following the addition of 50 μL of stop solution, the activities of IL-6, TNF-α, SOD, and GPx were estimated at 450 nm using a VersaMax microplate reader (Molecular Devices).

### 2.8. Protein Assay

To measure the amount of protein from cultured astrocytes, a BCA (bicinchoninic acid) Protein Assay Kit (TAKARA BIO, Inc., Nojihigashi, Japan) was used according to the manufacturer’s instructions. Supernatant was obtained from sonicated and centrifuged cells, and 100 μL of supernatant, standard, and working solution were added to a 96-well microplate and incubated at 60 °C for 1 h. After 1 h, the amount of protein was measured at 562 nm using a VersaMax microplate reader (Molecular Devices).

### 2.9. PI3K/MAPK Dual Pathway Activation Assay

To analyze activation of the PI3K/MAPK dual pathway, the Muse PI3K/MAPK Dual Pathway Activation Kit (Merck-Millipore, Darmstadt, Germany) was used according to the manufacturer’s instructions. The cultured astrocytes were grown in 24-well plates and exposed to ZnO NPs for 12 h. Then, the cells were deactivated by replacing the media with fresh media. Next, the cells were harvested using a cell scraper and centrifuged at 300× *g* for 5 min at 4 °C. The supernatant was discarded, and the cells were resuspended with 50 μL 1× assay buffer. Then, an equal volume of fixation buffer was added to the cell suspension and incubated for 10 min on ice. The cell suspension was centrifuged at 300× *g* for 5 min, and the supernatant was removed. To permeabilize the cells, 100 μL of ice-cold 1× permeabilization buffer was added and incubated for 10 min on ice. Afterwards, the cells were centrifuged at 300× *g* for 5 min, and the supernatant was removed. The cells were resuspended by adding a 90 μL 1× assay buffer and 10 μL of working antibody cocktail solution. Following incubation for 30 min in the dark at room temperature, 100 μL 1× assay buffer was added to the cells. Next, the cells were centrifuged at 300× *g* for 5 min, and the supernatant was removed. Finally, the cells were resuspended in 200 μL 1× Assay Buffer and analyzed using the Muse^TM^ Cell Analyzer.

### 2.10. Caspase-3/7 Assay

To assess caspase-3/7 activation, the Muse^®^ Caspase-3/7 Kit (Merck-Millipore) was used according to the manufacturer’s instructions. The cultured astrocytes were grown in 24-well plates and exposed to ZnO NPs for 12 h. Then, the cells were washed once with 1× HBSS and harvested using a cell scraper. The harvested cells were transferred to a Muse Assays microtube and centrifuged at 300× *g* for 5 min at 4 °C. The supernatant was removed, and each sample was resuspended with 50 μL 1× assay buffer. Next, 5 μL of Muse^TM^ Caspase-3/7 Reagent working solution was added to each sample and incubated at 37 °C in a 5% CO_2_ incubator for 30 min. After incubation, 150 μL of Muse^TM^ Caspase 7-AAD working solution was added to each sample, and the samples were vortexed at medium speed for 3 to 5 s. Then, the samples were incubated at room temperature for 5 min in the dark. Samples were assayed using the Muse^TM^ Cell Analyzer.

### 2.11. DAPI Staining

For fluorescent staining of DNA damage (intact and condensed or fragmented DNA), the medium of cultured astrocyte cells was aspirated, and the cells were washed three times with 0.1 M of PBS. Cells were fixed for 10 min in 4% paraformaldehyde and rinsed three times with the same buffer. Cells were permeabilized with 0.2% Triton X-100 for 5 min. After washing three times with PBS, cells were incubated at room temperature in DAPI labeling solution (1 µg/mL in PBS). Stained images were photographed using the fluorescence detector of an Olympus inverted microscope (Olympus, Tokyo, Japan).

### 2.12. Statistical Analysis

All the statistical analyses were conducted with the SAS 9.4 program using one-way analysis of variance (ANOVA) tests and Tukey’s multiple comparison test. *p* values < 0.05 were considered to indicate statistical significance, and all the experimental data are expressed as a mean ± standard error of the mean (SEM). 

## 3. Results

### 3.1. Nanoparticle Characterization

The ZnO NPs (NanoGard^®^) were characterized using field emission TEM (FETEM, Model: JEM2100F, JEOL Co., Tokyo, Japan; Figure 1A–C) with the accelerating condition set at 200 kV. An ultra-high resolution scanning electron microscope (UHR-SEM, Hitachi S-4800, Tokyo, Japan) at an accelerating voltage of 15 kV and 100,000× magnification was used to assess the ZnO NPs, which revealed rod-like shapes that were rectangular or hexagonal and agglomerized particles (Figure 1D). 

### 3.2. Dose-Dependent ZnO NP-Induced Toxicity in Neocortical Astrocyte Cultures 

Dose-dependent toxicity in the cultured astrocytes following treatment with ZnO NPs (1 µg/mL, 2 µg/mL, 3 µg/mL, 4 µg/mL, 5 µg/mL, and 10 µg/mL) was evaluated using LDH (Figure 2A) and MTT (Figure 2B) assays. Significant toxicity was observed following treatment with ZnO NPs at concentrations ≥3 µg/mL; thus, the pharmacological studies of test chemicals on ZnO NP-induced toxicity were conducted at a range between 3–10 µg/mL. The morphological findings of the control as well as treatment with 3 µg/mL and 10 µg/mL of ZnO NPs using phase-contrast microscopy are illustrated in Figure 2C–E. As the ZnO NP concentration increased, astrocytes displayed an abnormal morphology (Figure 2D,E) compared with the control (Figure 2C).

### 3.3. TEM Analyses of Astrocyte Morphology Following Exposure to ZnO NPs 

The astrocytes exhibited various types of cell morphology that contained intact organelles such as mitochondria, endoplasmic reticulum (ER), the nucleus, and lysosomes (Figure 3A). However, ZnO NPs induced the formation of autophagolysosomes as well as swelling of the ER (asterisk in Figure 3B); magnified images are shown in Figure 3C (arrow in Figure 3B) and Figure 3D (arrowhead in Figure 3B). The mitochondria and cytoplasmic and nuclear membranes maintained a generally intact state.

### 3.4. ZnO NP-Mediated Enhancement of Autophagy of Cultured Astrocytes

The exposure of cultured astrocytes to toxic concentrations of ZnO NPs (3 µg/mL, 5 µg/mL, and 10 µg/mL) significantly increased the number of autophagic LC3-positive cells at 12 h and 24 h compared with the control (Figure 4A). The threshold values (untreated) of autophagy intensity were 15.9 ± 2.4 (12 h) and 20.6 ± 1.0 (24 h). However, the mean autophagy intensity (MAI) in the control group was slightly decreased to 14.0 ± 0.6 at 12 h, but significantly increased to 25.1 ± 0.5 at 24 h. In cultured astrocyte cells exposed to each concentration (3 µg/mL, 5 µg/mL, and 10 µg/mL) of ZnO NPs, the MAIs at both 12 h and 24 h in the control groups were significantly increased by 152.9%, 147.1%, and 130.0% (12 h) and 131.5%, 156.2%, and 166.9% (24 h), respectively. Figure 4B,C present representative flow cytometry results of Muse autophagic LC3 detection at 12 h and 24 h following treatment with ZnO NPs. 

### 3.5. Effects of ZnO NPs on Caspase-3/7 Activity and DAPI Staining

The exposure of cultured astrocyte cells with ZnO NPs (3 µg/mL and 10 µg/mL) resulted in significantly increased caspase-3/7 activity at 12 h in a dose-dependent manner (Figure 5A). The total apoptotic population exhibited a twofold (9.1 ± 4.3% at 3 µg/mL) and threefold (15.5 ± 6.0% at 10 µg/mL) increase compared to the control (4.5 ± 3.4%). However, NAC completely blocked the ZnO NP-induced increase in caspase-3/7 activity compared with the control. Representative data obtained from flow cytometry is shown in Figure 5B. To identify the increase in apoptotic bodies, which is indicated by chromatin condensation or nuclear fragmentation, as induced by ZnO NPs, cultured astrocyte cells were stained with DAPI at 12 h after ZnO NP treatment. The control cells were almost intact, and no nuclear-condensed cells were observed (Figure 5C). However, the number of condensed apoptotic cell bodies was increased following treatment with 3 µg/mL and 10 µg/mL of ZnO NPs (Figure 5D,E). NAC treatment restored the control state (Figure 5F).

### 3.6. Effects of ZnO NPs on IL-6, TNF-α, SOD, and GPx Levels in Cultured Astrocyte Cells

Changes in the inflammatory cytokines IL-6 and TNF-α induced by ZnO NP exposure were evaluated in the media of cultured astrocyte cells. ZnO NPs significantly elevated the levels of IL-6 and TNF-α at 10 µg/mL (Figure 6A,B), while treatment with NAC completely blocked the increase in IL-6 and TNF-α levels induced by ZnO NPs (Figure 6A,B). By contrast, SOD and GPx levels were significantly decreased following treatment with 3 µg/mL and 10 µg/mL ZnO NPs (Figure 6C,D). NAC treatment was able to block the decrease in SOD at 10 µg/mL ZnO NPs (Figure 6C). However, NAC exhibited a significant protective effect following treatment with 3 µg/mL of ZnO NPs (Figure 6D).

### 3.7. Effect of ZnO NPs on PI3K/MAPK Activation in Cultured Astrocytes

PI3K/MAPK activity induced by ZnO NP exposure was examined to identify the relationship with apoptosis by flow cytometry in cultured astrocyte cells. ZnO NPs slightly but significantly induced activation of the dual pathway or MAPK in the PI3K/MAPK assay at 3 µg/mL and 10 µg/mL (Figure 7A,B). However, these concentrations of ZnO NPs did not greatly impact PI3K activity in cultured astrocytes. NAC treatment significantly inhibited the ZnO NP-induced activation of the dual pathway and MAPK. Representative flow cytometry data is shown in Figure 7C.

### 3.8. Effects of Meloxicam, Esculetin, and Phenidone on ZnO NP-Induced Toxicity 

Meloxicam, which is a selective COX-2 inhibitor, attenuated astrocyte injury in a concentration-dependent manner and exhibited a significant difference at 500 µM (Figure 8A). Phenidone (1 mM), a dual inhibitor of COX-2 and LOX, also significantly inhibited cell injury in ZnO NP-induced toxicity (Figure 8B). Based on LDH measurements, esculetin, which is a LOX inhibitor, significantly ameliorated ZnO NP-mediated astrocyte damage at concentrations of 30 µM and 60 µM (Figure 8C). An MTT reduction test also revealed protective effects of esculetin at concentrations of 10 µM and 30 µM (Figure 8D).

### 3.9. Effects of BML-257 and Rapamycin on ZnO NP-Induced Toxicity

The Akt inhibitor BML-257 significantly decreased ZnO NP-induced toxicity at 50 µM (Figure 9A). Rapamycin, a strong immunosuppressant, also significantly alleviated astrocyte damage at concentrations between 5–20 µM (Figure 9B). By contrast, at 30 µM of rapamycin, astrocyte toxicity was potentiated (Figure 9B).

### 3.10. Effects of Calcium Modulators, Antioxidants, and Metal Chelators on ZnO NP-Induced Toxicity

BAPTA/AM, which is an intracellular calcium chelator, significantly blocked ZnO NP-induced toxicity at a concentration range of 0.3 to 3 µM (Figure 10A). Nimodipine, which is an L-type calcium channel blocker, only ameliorated ZnO NP-induced toxicity at 100 µM (Figure 10B). Treatment with the antioxidant α-tocopherol (100 µM and 200 µM) contributed to the survival of astrocyte cells in ZnO NP-induced toxicity (Figure 10C), and NAC, which is a glutathione precursor, also potently protected cultured astrocyte cells in a concentration-dependent manner at 0.1 mM to 1.0 mM (Figure 10D). Metal chelators, such as TPEN (1 µM and 3 µM), a zinc chelator, and deferoxamine (100 µM and 300 µM), an iron chelator, significantly decreased the ZnO NP-induced damage of astrocyte cells in a dose-dependent manner (Figure 10E,F). A possible hypothesis based on our results is illustrated in Figure 11.

## 4. Discussion

The increased use of ZnO NPs can result in their consistent accumulation in the body [31], and they have been recognized as potential toxicants that can invade important internal organs through diverse routes such as the skin, respiration, food ingestion, and during medical treatment [31,32]. The brain is the most important organ in the control of the systemic body, and zinc is an endogenous metal that is distributed in hippocampal mossy fibers at micromolar concentrations, and can contribute to neurodegenerative diseases such as ischemic stroke and Alzheimer’s disease [33,34,35]. 

In this study, we determined that ZnO NPs were toxic at concentrations ≥3 µg/mL. The LDH and MTT assays showed similar results. Above this concentration, morphological cell damage was evident. Interestingly, we found that the mouse primary cultured astrocytes used in this study had 2.6-fold and 20-fold higher vulnerability to ZnO NPs compared to cultured rat primary astrocytes in previous studies by Wang et al. [13] and Sudhakaran et al. [14], respectively. Sruthi and Mohanan [23] found that ZnO NPs induce significant changes at concentrations ≥5 µg/mL, based on a MTT assay and analyses of the generation of reactive oxygen species (ROS) in a C6 glial cell line. Sharma et al. [36] reported that the 24-h inhibitory concentration 50 (IC_50_) of ZnO NPs is 6.6 µg/mL in microglial cells. Although the culture conditions in each of the above-mentioned studies differed somewhat, together, the results suggest that microglia have a twofold lower vulnerability than astrocytes [36]. Thus, astrocytes likely represent the first target when toxic levels of ZnO NPs pass through the BBB. Furthermore, this initial damage could trigger reactive astrogliosis and propagate secondary neuronal injuries. By contrast, Dineley et al. [37] reported that astrocytes are more resistant to zinc-mediated toxicity than neurons, because they have a greater buffering capacity for intracellular zinc ions and a different expression pattern of metallothionein isoforms (e.g., MT-1 and MT-II). However, in this study, astrocytes showed more vulnerability to ZnO NP toxicity than human SH-SY5Y neuroblastoma cells.

Autophagy is the elimination of metabolic debris to maintain cell survival. Autophagic functions lead to abnormal deterioration during the acceleration of the aging process and in neurodegenerative states [38]. In the present study, ZnO NPs increased mean autophagy intensity in a concentration-dependent manner at 12 h and 24 h. These results are partly consistent with those of previous reports showing that ZnO NP treatment increased autophagic biomarkers, such as LC3 expression and LC3-FITC antibody-positive cells in ovarian cancer cells and macrophages [39,40]. Lee et al. [41] also reported that the zinc ions can participate in oxidative injury-mediated autophagy. 

Using TEM, we confirmed that autophagolysosome formation and ER swelling were evident following treatment with ZnO NPs. These results suggest that ZnO NPs may induce the production of pro-inflammatory cytokines, such as IL-6 and TNF-α, via activation of the ER stress–autophagy axis [42].

In primary astrocyte cultures, treatment with ZnO NPs increased apoptosis (twofold and threefold increases compared to controls) dose-dependently at 3 µg/mL and 10 µg/mL at 12 h, respectively. However, NAC reversed this effect to control levels. Sudahkaran et al. [14] reported that ZnO NPs significantly increased caspase 3/7 activity dose-dependently at 5 to 80 µg/mL. Wang et al. [13] reported that treatment with ZnO NPs resulted in caspase 3/7 activity only at 12 µg/mL, and not at 4 µg/mL and 8 µg/mL, between 6–24 h. This inconsistency in caspase 3/7-mediated apoptosis following treatment with ZnO NPs may be a result of differences in culture conditions or the cellular environment. As further confirmation of apoptotic findings, nuclear condensation, as observed by DAPI staining, was increased dose-dependently following treatment with 3 µg/mL and 10 µg/mL of ZnO NPs in parallel with ROS production. Here, ZnO NP-mediated nuclear condensation was diminished following treatment with the general antioxidant NAC. These results suggest that ROS play a pivotal role in caspase-3/7 activation and DAPI staining induced by ZnO NPs.

Recently, in vivo studies have shown that levels of the pro-inflammatory cytokines IL-6 and TNF-α are elevated following treatment with ZnO NPs, whereas those of SOD and GPx are reduced [12]. Roy et al. [40] reported that ZnO NPs decrease GPx, glutathione reductase, catalase, and glutathione-S-transferase levels, as well as ROS production in macrophages. In our study, IL-6 and TNF-α were elevated following treatment with 10 µg/mL of ZnO NPs, but not with 3 µg/mL. However, the levels of endogenous antioxidant enzymes SOD and GPx were significantly decreased following treatment with 3 µg/mL and 10 µg/mL ZnO NPs. These results suggest that both the elevation of oxidative stress and release of pro-inflammatory cytokines may occur following treatment with ZnO NPs at concentrations greater than 10 µg/mL in astrocyte cultures aged 3–5 weeks. NAC treatment contributed to the recovery of these pathological changes. Interestingly, a twofold increase in TNF-α was observed following treatment with 3 µg/mL of ZnO NPs in aged astrocyte cultures (24 weeks; data not shown). These results suggest that aged astrocytes are more vulnerable to ZnO NP-induced pro-inflammatory reactions than young astrocyte cells.

Recently, PI3K/Akt and MAPK have been shown to be involved in autophagy or apoptosis and cell survival through various cellular stress mechanisms or pro-inflammatory cytokines in ZnO NP toxicity [40,43]. In those studies, ZnO NP treatment activated the PI3K/Akt and MAPK dual signaling pathway in macrophages. Similarly, we also found that ZnO NPs induced significant elevation of PI3K/MAPK dual activation in primary cultured astrocytes. The antioxidant NAC and the PI3K inhibitor LY294002 reduced the activation of this pathway. These results suggest that oxidative stress—which is a result of the loss of activity of antioxidant enzymes such as SOD and GPx—following ZnO NP exposure may play a pivotal role in astrocyte injury.

In a previous study, ZnO NPs were introduced to elevate the expression of COX-2 as well as pro-inflammatory cytokines in macrophages [43]. We were interested in the roles of COX-2 or LOX in ZnO NP toxicity in different types of cultures. Thus, we evaluated the protective ability of COX-2 or LOX inhibitors in ZnO NP-induced toxicity in primary astrocyte cultures. Inhibitors of COX-2 or LOX, including a dual inhibitor, decreased astrocyte injury. These results are different from a previous study that used human dopaminergic SH-SY5Y neuroblastoma cells. Thus, it is important to select suitable anti-inflammatory drugs in ZnO NP-related inflammatory diseases.

We evaluated the effects of the Akt inhibitor BML-257 in ZnO NP-induced astrocyte injury. BML-257 attenuated astrocyte injury induced by ZnO NPs, suggesting that the Akt signaling activation process, rather than the PI3K process, may contribute to astrocyte injury. In contrast with previous studies that used neuroblastomas, the PI3K inhibitor LY294002 did not significantly inhibit ZnO NP-induced astrocyte toxicity in LDH assays (data not shown) [19]. Roy et al. [40] reported that ZnO NPs induce autophagy and apoptosis via inhibition of the PI3K/Akt/mTOR signaling process in macrophages. Rapamycin is a well-known immunosuppressant and inhibitor of mTOR that results in increased autophagy [44]. A recent study found that rapamycin ameliorates seizure-induced astrocyte injury [45] and rescues astrocytes and neurons by modulating the inflammatory response and elevations of p-Akt [46]. However, the results of the present study revealed that rapamycin exhibited biphasic effects such that it rescued astrocytes from ZnO NP-induced injury at concentrations from 5 to 20 µM, but exacerbated astrocyte injury at 30 µM. Although rapamycin may be a potential therapeutic strategy for the treatment of astrocyte injury, its risk of toxicity cannot be ignored. 

Maintaining an appropriate balance in intracellular calcium levels is also very important for the survival of astrocytes. In the present study, BAPTA/AM, which is an intracellular calcium chelator and L-type calcium channel blocker, reduced ZnO NP-induced damage to astrocytes. These results suggest that the excessive accumulation of intracellular zinc ions resulting from the permeabilization of dissolved ZnO NPs in cultured astrocytes may trigger calcium dyshomeostasis [19,47]. However, calcium overload can deplete glutathione contents in astrocytes [47]. When the antioxidants NAC and α-tocopherol were applied to the cultured astrocytes, cell viability recovered to a normal state. These results suggest that oxidative damage due to the loss of antioxidant enzymes, such as GPx and SOD may be a crucial factor in ZnO NP-induced toxicity in astrocytes. Additionally, TPEN, which is a membrane-permeable specific zinc chelator, and the iron chelator deferoxamine both ameliorated ZnO NP-induced toxicity in cultured astrocytes in concentration-dependent manners. Deferoxamine inhibits buthionine sulfoximine-induced glutathione depletion and zinc-induced lipid peroxidation [48,49] Accordingly, it might contribute to the protection of astrocytes in ZnO NP-mediated oxidative injury. In summary, we found that ZnO NPs may induce autophagy partly via the PI3K/Akt/mTOR pathway and caspase-3/7-mediated apoptosis via the increased release of IL-6 and TNF-α, the depletion of SOD and GPx, and increased oxidative stress by zinc/calcium dyshomeostasis in primary cultured astrocytes.

## Figures and Tables

**Figure 1 nanomaterials-09-01043-f001:**
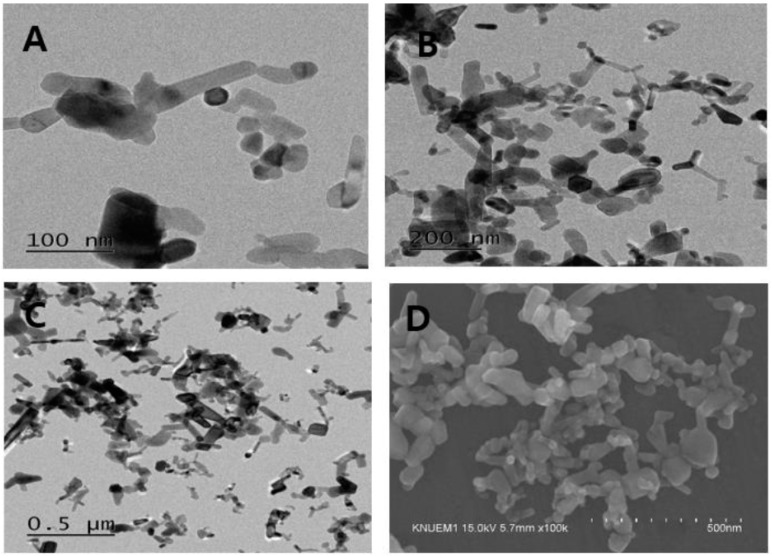
TEM (**A**–**C**) and scanning electron microscopy (**D**) images of zinc oxide nanoparticles (ZnO NPs). ZnO NPs exhibited an agglomerated and rod-shape morphology.

**Figure 2 nanomaterials-09-01043-f002:**
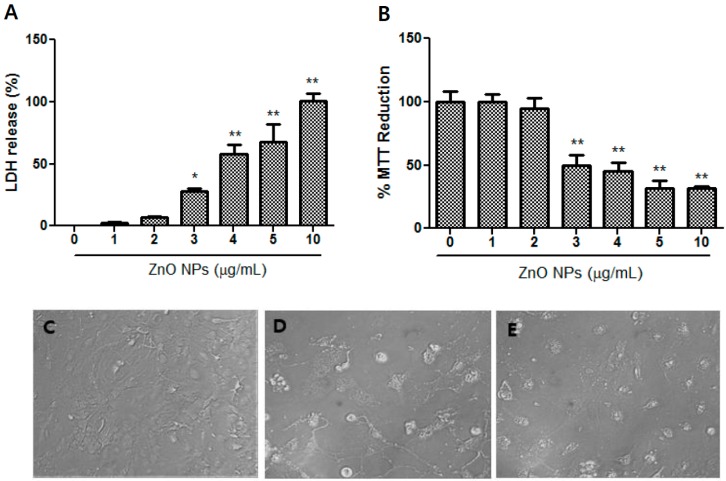
Dose-toxicity of ZnO NPs in primary cultured astrocyte cells. Cell viability was assessed using lactate dehydrogenase (LDH) release (**A**) and 3-(4,5-dimethylthiazol-2-yl)-2,5-diphenyltetrazolium bromide (MTT) reduction (**B**) assays. Representative microscopic images of astrocytes using phase-contrast (**C**–**E**): (**C**) control; (**D**) 3 µg/mL of ZnO NPs; (**E**) 10 µg/mL of ZnO NPs. Data represents the mean ± SEM. * *p* < 0.05, ** *p* < 0.01 compared to control (*n* = 4).

**Figure 3 nanomaterials-09-01043-f003:**
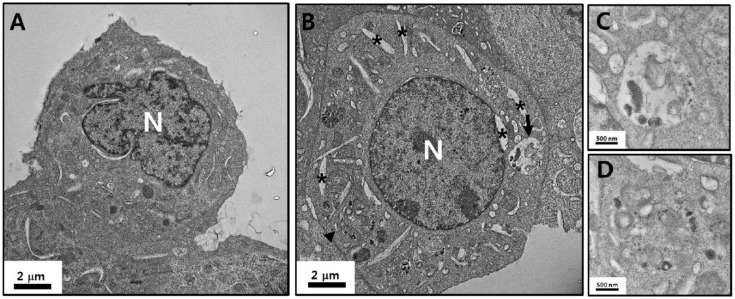
TEM images of astrocyte morphology. TEM images of astrocytes treated with phosphate-buffered saline (PBS) (**A**) and astrocytes incubated with ZnO NPs at a concentration of 3 µg/mL for 24 h (**B**). (**C**,**D**) are magnified images of the arrow and arrowhead in (**B**), respectively. The black arrow and arrowhead indicate autophagolysosomes induced by ZnO NPs. * Swelling of the endoplastic reticulum (ER). N: nucleus.

**Figure 4 nanomaterials-09-01043-f004:**
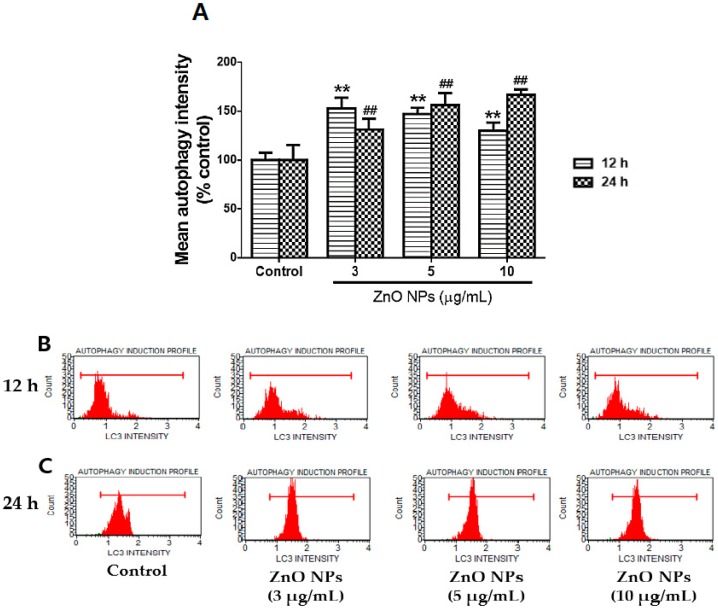
Effects of ZnO NPs on autophagic LC3 intensity in primary cultured astrocyte cells. Mean autophagy intensity gradually increased at toxic concentrations of ZnO NPs at 12 h and 24 h (**A**). Representative flow cytometry data for the control and ZnO NPs (3 µg/mL, 5 µg/mL, and 10 µg/mL) at 12 h (**B**) and 24 h (**C**). Data represents the mean ± SEM. ** *p* < 0.01 and ## *p* < 0.01 compared to the respective controls (*n* = 4).

**Figure 5 nanomaterials-09-01043-f005:**
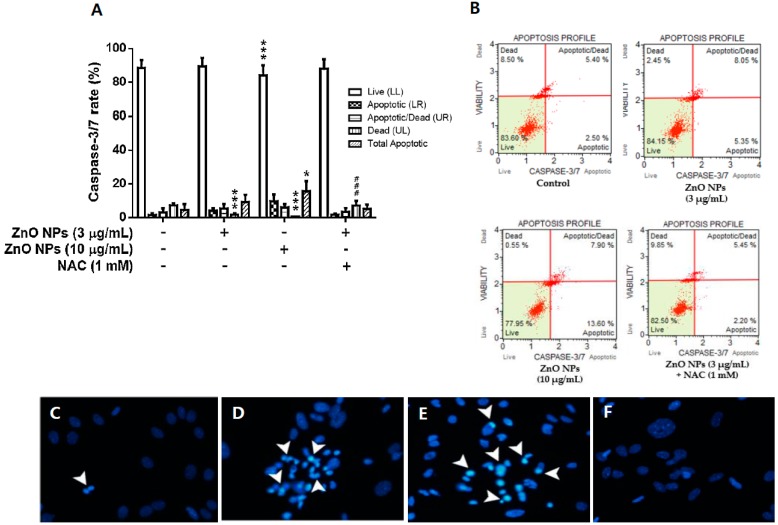
Effects of ZnO NPs on apoptosis profiles (live, apoptotic, apoptotic/dead, dead, and total apoptotic) of primary cultured astrocyte cells. (**A**). Representative flow cytometry data for the control, ZnO NPs (3 µg/mL and 10 µg/mL), and ZnO NPs (3 µg/mL) + N-acylcysteine (NAC) at 12 h are illustrated in (**B**). Nuclear condensation observed by 4′,6′-diamidino-2-phenylindole (DAPI) staining in untreated cultured astrocyte cells (**C**) and cultured astrocyte cells treated with 3 µg/mL of ZnO NPs (**D**), 10 µg/mL of ZnO NPs (**E**), and 3 µg/mL of ZnO NPs + NAC (1 mM) (**F**) for 24 h. Arrowhead indicates fluorescent DAPI–DNA complex (+) cells. Data represents the mean ± SEM. ** *p* < 0.01 and ## *p* < 0.01 compared to the respective controls (*n* = 4).

**Figure 6 nanomaterials-09-01043-f006:**
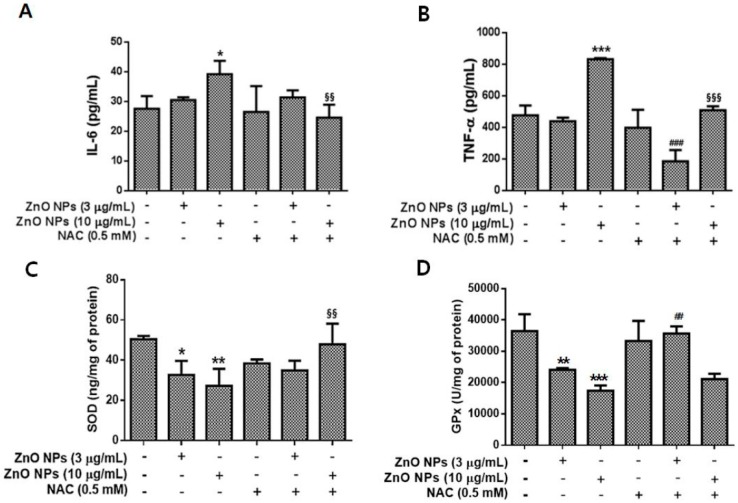
Effects of ZnO NPs on the pro-inflammatory cytokines interleukin (IL)-6 (**A**) and tumor necrosis factor (TNF)-α (**B**) and the antioxidant enzymes superoxide dismutase (SOD) (**C**) and glutathione peroxidase (GPx) (**D**) in primary cultured astrocyte cells. The levels of IL-6 and TNF-α were significantly elevated at 24 h following exposure to 10 µg/mL of ZnO NPs compared with the control. NAC treatment (0.5 mM) restored the levels of IL-6 and TNF-α to that of the control. The levels of SOD and GPx were significantly decreased by ZnO NP treatment in a dose-dependent manner. Data represents the mean ± SEM. * *p* < 0.05, ** *p* < 0.01, *** *p* < 0.001 compared to the respective controls. ## *p* < 0.01 compared to 3 µg/mL ZnO NPs; ^§§^
*p* < 0.01, ^§§§^
*p* < 0.001 compared to 10 µg/mL of ZnO NPs (*n* = 4).

**Figure 7 nanomaterials-09-01043-f007:**
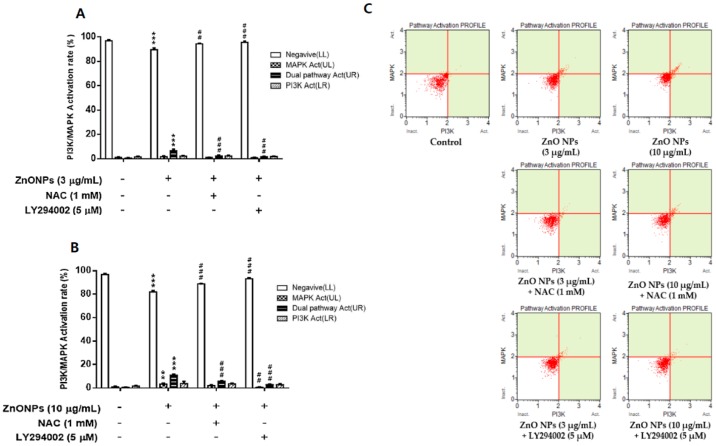
Effect of ZnO NPs on activation of the phosphatidylinositol 3-kinase (PI3K)/ mitogen-activated protein kinase (MAPK) dual pathway in primary cultured astrocyte cells. ZnO NPs significantly increased PI3K/MAPK dual pathway activation at 3 µg/mL (**A**) and 10 µg/mL (**B**). NAC and LY294002 significantly inhibited PI3K/MAPK dual pathway activation. Representative flow cytometry data for the control, ZnO NPs (3 and 10 µg/mL), ZnO NPs (3 or 10 µg/mL) + NAC, and ZnO NPs (3 or 10 µg/mL) + LY294002 (5 µM) at 12 h are illustrated in (**C**). Data represents the mean ± SEM. ** *p* < 0.01, *** *p* < 0.001 compared to the respective controls. ## *p* < 0.01, ### *p* < 0.001 compared to 3 or 10 µg/mL ZnO NPs (*n* = 4).

**Figure 8 nanomaterials-09-01043-f008:**
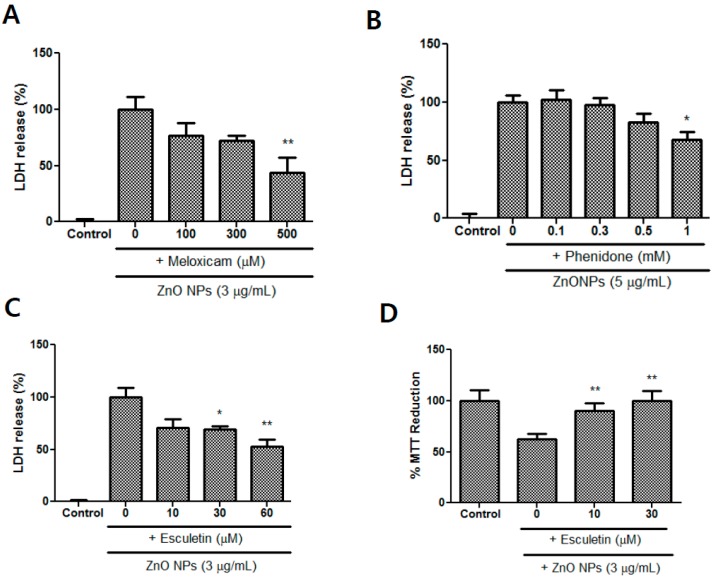
Protective effects of meloxicam (**A**), phenidone (**B**), and esculetin (**C**,**D**) against ZnO NP-induced toxicity in primary astrocyte cultures. Cytotoxicity was evaluated using LDH release (**A**–**C**) and MTT reduction (**D**) assays. Data represents the mean ± SEM. * *p* < 0.05, ** *p* < 0.01 compared to control (*n* = 4).

**Figure 9 nanomaterials-09-01043-f009:**
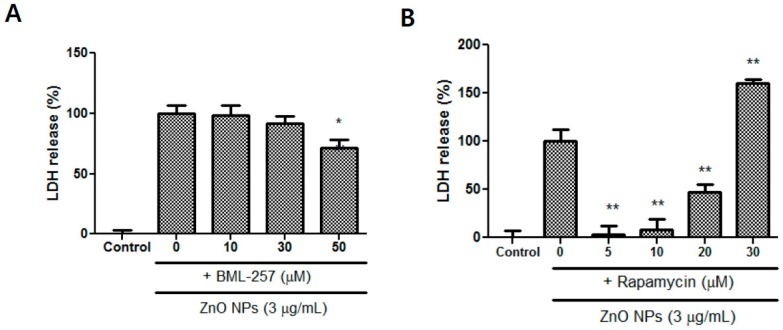
Protective effects of BML-257 (**A**) and rapamycin (**B**) on ZnO NP-induced toxicity in primary astrocyte cultures. At a higher dose (30 µM) of rapamycin, ZnO NP-mediated toxicity was potentiated. Cytotoxicity was evaluated using an LDH release assay. Data represents the mean ± SEM. * *p* < 0.05, ** *p* < 0.01 compared to ZnO NPs (*n* = 4).

**Figure 10 nanomaterials-09-01043-f010:**
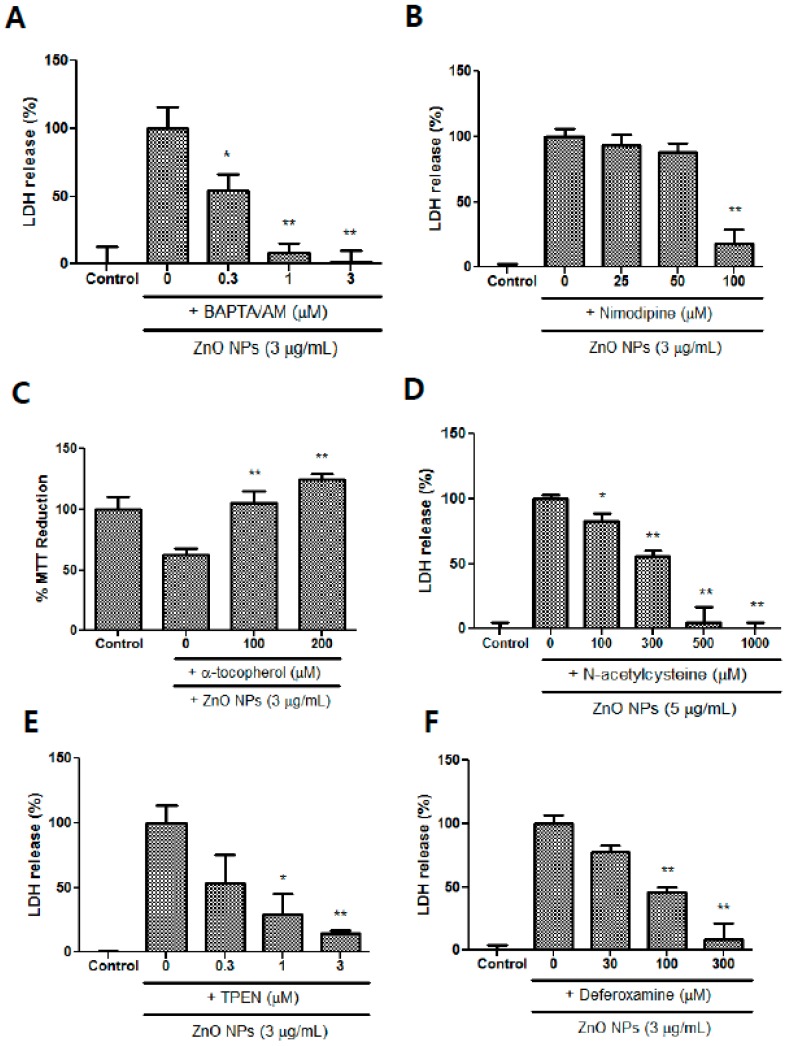
Protective effects of BAPTA/AM (1,2-bis(2-aminophenoxy)ethane-N,N,N′,N′-tetraacetic acid/acetoxymethyl ester) (**A**), nimodipine (**B**), α-tocopherol (**C**), NAC (**D**), N,N,N′,N′-tetrakis-(2-pyridylmethyl) ethylenediamine (TPEN) (**E**), and deferoxamine (**F**) on ZnO NP-induced toxicity in astrocyte cultures. Cytotoxicity was evaluated using LDH release and MTT reduction assays. Data represents the mean ± SEM. * *p* < 0.05, ** *p* < 0.01 compared to ZnO NPs (*n* = 4).

**Figure 11 nanomaterials-09-01043-f011:**
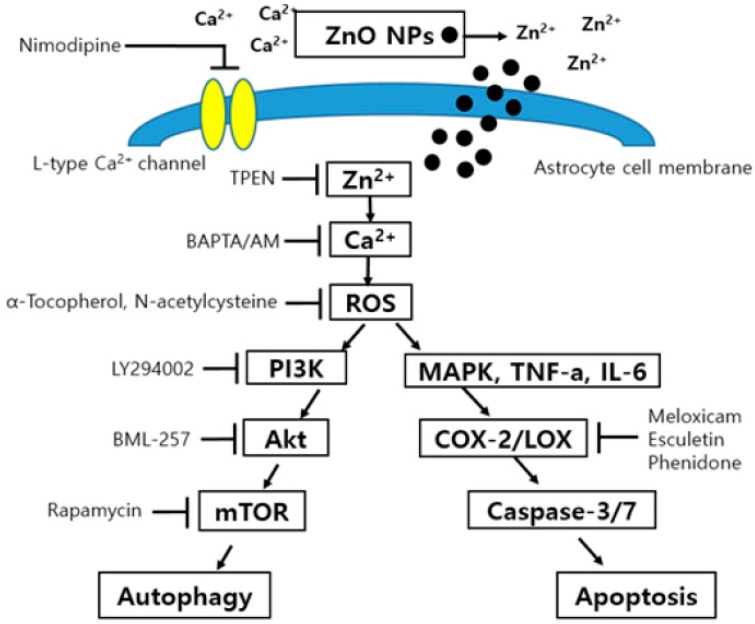
Possible mechanism of ZnO NP-induced toxicity in primary cultured astrocyte cells.

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
