# Peer review of "Zinc Oxide Nanoparticles Induce Autophagy and Apoptosis via Oxidative Injury and Pro-Inflammatory Cytokines in Primary Astrocyte Cultures"

_nanomaterials, 2019, doi:10.3390/nano9071043_

Reviewer 1 Report

P { margin-bottom: 0.21cm; }A:link {  }

I have appreciated the efforts carried out by authors during the revision (especially concerning the statistical analysis). Anyway, some minor points needs to be addressed by the reviewers:

-Abstract, Line 26. “Calcium-modulating compounds, antioxidants, and zinc/iron chelators also ameliorated ZnO NP-induced toxicity.” What does it mean “ameliorated toxicity”? Probably the authors mean: “Calcium-modulating compounds, antioxidants, and zinc/iron chelators decreased the ZnO NP-induced toxicity”. Please, double check the english mistakes and typos along the manuscript.

-Line 220. Authors described as “Astrocyte cells were visibly shrunken or demonstrated loss of cell bodies”, please consider that you should refer to “atrocytes”, or to “astrocyticglial cells”, but not to “astrocyte cells”; moreover, “shrunken” is not a appropriate scientific description of cell morphology; finally, you do not demonstrate the loss of cell body! Please, substitute this sentence with “Astrocytes displayed an abnormal morphology” or with a similar one.

-Line 239. Please substitute “enhancemen” with “enhancement” in the title

-Concerning the paper references that the author was not able to find in Pubmed, you can find them at the following links (https://www.sciencedirect.com/science/article/pii/S1748013216304121;https://pubs.acs.org/doi/abs/10.1021/acsami.7b04323).

Author Response

Response to the Reviewer’s comments

I have appreciated the efforts carried out by authors during the revision (especially concerning the statistical analysis). Anyway, some minor points need to be addressed by the reviewers:

Reviewer(s)’ Comments to Author:

[Reviewer #1]

Manuscript Ref. No.: Nanomaterials-553685

Manuscript title: Zinc Oxide Nanoparticles Induce Autophagy and Apoptosis via Oxidative Injury and Pro-inflammatory Cytokines in Primary Astrocyte Cultures

Comments.

-Abstract, Line 26. “Calcium-modulating compounds, antioxidants, and zinc/iron chelators also ameliorated ZnO NP-induced toxicity. “What does it mean “ameliorated toxicity”? Probably the authors mean: “Calcium-modulating compounds, antioxidants, and zinc/iron chelators decreased the ZnO NP-induced toxicity”. Please, double check the English mistakes and typos along the manuscript.

Response: In response to this comment, we have changed “ameliorated” to “decreased” in abstract, Line 26 including discussion, Line 419 and double checked the English mistakes and typos along the manuscript. According to reviewer’s comment, we changed “astrocyte cells” into “astrocytes” in page 14, line 411.

-Line 220. Authors described as “Astrocyte cells were visibly shrunken or demonstrated loss of cell bodies”, please consider that you should refer to “astrocyte”, or to “astrocyticglial cells”, but not to “astrocyte cells”; moreover, “shrunken” is not an appropriate scientific description of cell morphology; finally, you do not demonstrate the loss of cell body! Please, substitute this sentence with “Astrocytes displayed an abnormal morphology” or with a similar one. 

Response: In response to this comment, we changed “astrocyte cells were visibly shrunken or demonstared loss of cell bodies” into “astrocytes displayed an abnormal morphology” in page 6, line 220 in results. Similarly, we removed “such as cell shrinkage and loss of cell bodies” in page 13, line 355-356 according to reviewer’s comment.

-Line 239. Please substitute “enhancemen” with “enhancement” in the title

Response: In response to this comment, we have changed “enhancemen” into “enhancement” in line 239.

-Concerning the paper reference that the author was not able to find in Pubmed, you can find them at the following links

(https://www.sciencedirect.com/science/article/pii/s1748013216304121;https://pubs.acs.org/doi/abs/10.1021/acsami.7b04323)

Response: As your advice, we newly added two references as No. 3 and No. 4 in page 16, line 466-469. In addition, we changed “piezoelectric scaffolds for orthopedic applications” to “piezoelectric nanomaterials for biomedical applications” in page 1, line 34.

Reviewer 2 Report

the revised MS has improved, recommendation for publication. Thank you.

Author Response

[Reviewer #2]

The revised MS has improved, recommendation for publication.Thank you.

Response: We really thanks to reviewer’s generosity about nice comment and advice.

Reviewer 3 Report

The authors addressed almost all my concerns. They also included the suggested figure about the context of the inhibitors and how the inhibited components are involved in ZnO-mediated cell death. However, the figure is not sufficiently explained. Furthermore, I do not see sufficient experimental support for the order of Ca and ROS in the figure. Zn can also directly induce ROS and Ca can be increased due to apoptotic signaling - not only via direct Zn-interaction. 

Author Response

[Reviewer #3]

The authors addressed almost all my concerns. They also included the suggested figure about the context of the inhibitors and how the inhibited components are involved in ZnO-mediated cell death. However, the figure is not sufficiently explained. Furthermore, I do not see sufficient experimental support for the order of Ca and ROS in the figure. Zn can also directly induce ROS and Ca can be increased due to apoptotic signaling-not only via direct Zn-interaction.

Response: In possible mechanism of ZnO NPs as illustrated by Fig. 11, comments by reviewer 3 is absolutely right. We could not suggest direct results on elevation of ROS and intracellular calcium levels induced by ZnO NPs exposure. Instead, we suggested pharmacological data using antioxidant or inhibitor of ROS production, N-acetylcysteine and intracellular calcium chelator, BAPTA/AM or L-type calcium channel blocker, nimodipine. However, in our previous study using human dopaminergic neuroblastoma SH-SY5Y cell line, we have directly identified a significant increase of ROS induced by exposure of ZnO NPs using 2’,7’-dichlorofluorescin diacetate (DCFH-DA) to dichlorofluorescin (DCF) conversion assay method. Here, N-acetylcysteine (NAC) significantly inhibited ZnO NP-induced elevation of ROS. These results consistent with present data. We also have plentiful experiences about excitotoxicity research using in vitro ischemia (by oxygen/glucose deprivation) models using primary neuron or astroglial cultures. In this condition, acute accumulation of [Ca2+]i and [Zn2+]i can trigger ROS production (Clausen et al., 2013) in neuronal injury. According to Huang et al, ZnO NPs exposure increased intracellular calcium levels, [Ca2+]i in a concentration- and time-dependent manner which was attenuated by NAC. Here, nifedipine, calcium channel blocker, partly attenuated the increase of [Ca2+]i (Huang et al., 2010). This means that extracellular calcium could be move into inside of the cell in ZnO NP toxicity. As you understand, the gradient gap of free calcium concentration is 10,000 times between intracellular and outcellular calcium concentration. Accordingly, free calcium in outcellular space always easy to invade into intracellular space through several routes (calcium or NMDA channels etc.) by various toxic states. In our ELISA study, endogenous antioxidant enzymes, such as SOD and GPx decreased by exposure of ZnO NPs. This means we could anticipate deprivation of SOD or GPx can trigger ROS production. However, it is not easy to compare the pivotal role on interrelationship between ROS and two cationic ions in toxic state induced by ZnO NPs. To clarify the direct evidence on change of calcium and ROS, if situation will be permitted, we are trying to measure the changes of free calcium levels using confocal microscope when ZnO NPs were exposed to neuronal or astroglial cells. Especially, we would like to give real thanks about sharp comment by reviewer 3.

Round  2

Reviewer 1 Report

Authors have replied to all the comments of the reviewers. Manuscript is now accepted for publication.

This manuscript is a resubmission of an earlier submission. The following is a list of the peer review reports and author responses from that submission.

Round  1

Reviewer 1 Report

In the manuscript "Zinc Oxide Nanoparticles Induce COX-2- and LOX-mediated Oxidative Injury and Akt/mTOR-related Autophagic Cell Death in Primary Astrocyte Cultures", authors investigated the potential toxic concentrations of ZnO NPs and associated autophagy-related injuries in primary neocortical astrocyte cultures. The work is scientifically interesting but the following points need to be addressed by authors:

- Why the optical microscope images of Figure 2, 5 , 6, 7, 8 and 9 have a green or red background?! It seems that filter cubes were applied to the optical path of the microscope. The optical images should be presented without this background and the contour of the cells should be visible.

- Moreover, the description of the optical images in Figure 2 is completely missing. It is only reported that "procedures for morphological confirmation using phase-contrast microscopy(Figs. 2C–D) and for viability matching with the LDH results using trypan blue staining (Figs. 2E–H) were performed."

Please describe the difference between Fig 2C and D, as well as between Fig 2E, F, G and H.

- Concerning all the experiments with trypan blue, it would be also appreciated to report the percentage of trypan blue positive cells for each experimental class.

- In Figure 4C, please report the thresholded LC3 intensities with the same red colour used for Figures 4 D and E (and not in blue). Moreover, indicate in the text what the selected LC3 intensities refer to.

- The Introduction is quite bare and not very appealing. Please, include in the Introduction the most interesting applications of Zinc oxide nanomaterials in nanomedicine. Include at least photodynamic therapy and wireless piezoelectric stimulation. For these applications, add the following references:

photodynamic therapy (DOI 1: 10.3390/nano8030143; DOI 2: 10.1088/1612-2011/11/2/025601) 

wireless piezoelectric stimulation (DOI 3:  DOI 4: 10.1016/j.nantod.2016.12.005; DOI 4: 10.1021/acsami.7b04323)

Reviewer 2 Report

Comments to the Author

Manuscript „Zinc Oxide Nanoparticles Induce COX-2- and LOX mediated Oxidative Injury and Akt/mTOR-related  Autophagic Cell Death in Primary Astrocyte Cultures”

For a better understanding of the reported results, the authors should take the comments below into consideration.

1.       The aim of the paper is not clear, especially the reason for performing these experiments. Are these various chemicals used with ZnO NPs to identify the signaling pathways, or, by calling them “rescuing chemicals”, to identify possibilities for cell protection or even treat cell injury by ZnO NPs?

2.       The concentrations of ZnO NPs is set with 3, 5 and 10 μg/ml. Is this a concentration that can be reached in vivo realistically? Are there situations, in which a high exposure to these particles is expected? Please add references.

3.       To measure cell damage by ZnO NPs, the authors compare groups of astrocytes exposed to ZnO NPs alone and ZnO NPs with “rescuing chemicals”. It is not correct to use the term “rescuing chemicals” in this context, as there is not clear yet, which effect these substances will have on ZnO NPs induced cell damage.

4.       Over ten chemicals are tested in order to examine the effect on ZnO NP-induced cell damage. In the discussion, the results of only a few of them are interpreted. It is not clear, whether the results of the lacking agents were expected, or even, the function of every agent.

5.       Furthermore, the discussion lacks the explanation, why LOX and COX-2 could be blocked similarly, while Phenidone, an inhibitor of both COX-2 and LOX, failed to reduce cell damage. There should be included an attempt to interpret the findings in a pathological context as well as an interpretation in comparison to existent literature. 

6.       The authors talk about positive medical applications of ZnO NPs. The authors refer to a previous paper, in which the ZnO NP induced cell damage on neuroblastoma cells is examined. Therefore, investigating the toxic concentration of ZnO NPs in astrocytes, the authors should compare these concentrations in order to interpret the results in a clinical context. 

7.       The authors show the dose-dependent effect of ZnO NPs on astrocytes. It would be interesting to see, whether there is a time-dependence on these effects as well.

Overall, this is a detailed paper about autophagy in ZnO NP-exposed cells. Methods were selected thoroughly. Nevertheless, the authors should pay more attention to the characterization of the used chemicals and the aim of the paper. 

Reviewer 3 Report

This is a very nice study on the effect of ZnO nanoparticles on primary astrocytes. The experiments seem conclusive and convincing. I have a few minor concerns regarding the presentation: 

- the microscopy images are not of good quality. E.g. in Fig. 5 E (and others) I cannot see any cells. Please improve the quality or move the images into the SI. The LDH or MTT tests respectively show the results more convincingly. 

-the scale bar annotation in Fig. 3 C and D is not visible

- some minor annotation issues: in the text under 3.3 you refer to fig. 2B instead of 3B and the caption for Fig. 5H is missing (and possibly others, please check carefully!). 

- The choice of inhibitors seems rather random. The story line (in results and in discussion) is missing the context. Please add a figure that shows the signaling pathways and how LOX, mTOR, Ca, tocopherol etc. are connected. Then explain the Figure and the context and how they are related to yield a better picture of what ZnO is affecting in the cells. 

- Are the iron chelators specific for iron or would they also chelate Zn? Please specify in the text or explain why the amount of iron might matter in the toxicity of ZnO. Otherwise it is not understandable why the chelators have an effect.